# Kangaroo Mother Care implementation research to develop models for accelerating scale-up in India and Ethiopia: study protocol for an adequacy evaluation

KMC Scale-Up Study Group

**Correspondence to**
Dr Rajiv Bahl; bahlr@who.int

## ABSTRACT

**Introduction** Kangaroo Mother Care (KMC) is the practice of early, continuous and prolonged skin-to-skin contact between the mother and the baby with exclusive breastfeeding. Despite clear evidence of impact in improving survival and health outcomes among low birth weight infants, KMC coverage has remained low and implementation has been limited. Consequently, only a small fraction of newborns that could benefit from KMC receive it.

**Methods and analysis** This implementation research project aims to develop and evaluate district-level models for scaling up KMC in India and Ethiopia that can achieve high population coverage. The project includes formative research to identify barriers and contextual factors that affect implementation and utilisation of KMC and design scalable models to deliver KMC across the facility-community continuum. This will be followed by implementation and evaluation of these models in routine care settings, in an iterative fashion, with the aim of reaching a successful model for wider district, state and national-level scale-up. Implementation actions would happen at three levels: 'pre-KMC facility'—to maximise the number of newborns getting to a facility that provides KMC; 'KMC facility'—for initiation and maintenance of KMC; and 'post-KMC facility'—for continuation of KMC at home. Stable infants with birth weight<2000 g and born in the catchment population of the study KMC facilities would form the eligible population. The primary outcome will be coverage of KMC in the preceding 24 hours and will be measured at discharge from the KMC facility and 7 days after hospital discharge.

**Ethics and dissemination** Ethics approval was obtained in all the project sites, and centrally by the Research Ethics Review Committee at the WHO. Results of the project will be submitted to a peer-reviewed journal for publication, in addition to national and global level dissemination.

**Study status** WHO approved protocol: V.4—12 May 2016—Protocol ID: ERC 2716. Study implementation beginning: April 2017. Study end: expected March 2019.

**Trial registration number** Community Empowerment Laboratory, Uttar Pradesh, India (ISRCTN12286667); St John's National Academy of Health Sciences, Bangalore, India and Karnataka Health Promotion Trust, Bangalore, India (CTRI/2017/07/008988); Society for Applied Studies,

### Strengths and limitations of this study

► This will be the first implementation research project to develop and evaluate models for achieving high population coverage with Kangaroo Mother Care (KMC).

► The project is a partnership between research groups and local governments, implemented at large scale, covering populations in diverse regions in two countries, Ethiopia and India, of high prevalence of low birth weight.

► This study has a significant limitation, the lack of a concomitant comparison group. The models' success in increasing KMC coverage will be assessed through the comparison with data at the beginning of the project that indicated that KMC is a rare practice in these settings, covering under 5% of low birth weights, and that competing intervention promoting KMC is not currently implemented in study sites.

Delhi (NCT03098069); Oromia, Ethiopia (NCT03419416); Amhara, SNNPR and Tigray, Ethiopia (NCT03506698).

## INTRODUCTION

Globally, each year 15 million neonates are born preterm and consequently at a high risk of mortality.[1 2] South Asia and sub-Saharan Africa account for almost two-thirds of the world's preterm babies.[3] Kangaroo Mother Care (KMC) is a low-cost intervention involving early, continuous and prolonged skin-to-skin contact between mother and baby and exclusive breastfeeding.[4] KMC has been demonstrated to promote physiologic stability, a thermally supportive environment, reduce risk of serious infections and reduce the mortality among hospitalised, stable preterm and low birth weight (LBW) infants.[5] Despite the benefits, coverage of KMC has remained low and implementation has largely been limited to specialised hospitals.[6–8] As

a result, only a small fraction of newborns that could benefit from KMC receive it. A set of seven studies has been planned in Ethiopia and India to develop and test models of scaling-up KMC effectively.

## AIMS OF THE IMPLEMENTATION RESEARCH

**Aim:** To develop a delivery model that will result in high coverage (80%+) and quality of KMC for the target population.

### Additional aims
► To design and identify the most promising delivery model through formative research combined with an iterative quality improvement (QI) approach.
► To implement the most promising model in the study facilities and evaluate quality and coverage of KMC in the population they serve.

## METHODS/DESIGN

We are undertaking an international, multi-site, implementation study using a mixed-methods design that integrates formative and evaluative components into iterative improvement cycles. Our intervention promotes KMC for babies born <2000 g across the community-facility continuum.

### Population

The population will be newborns with birth weight under 2000 g born in the geographic area targeted by the intervention. It includes all babies born in health facilities and in the community. For those newborns who are severely sick according to predefined criteria (not tolerating oral feeds, severe respiratory distress including, respiratory rate less than 20 breaths per minute, grunting, central cyanosis, severe chest indrawing, convulsions, unconsciousness and severe hypothermia of less than 32°C), KMC will be delayed until they stabilised.

### Intervention

The intervention will be the provision of KMC, defined as skin-to-skin contact with the mother or a replacing caregiver, for as long as possible during day and night, with a minimum of at least 8 hours over a 24-hours period, along with exclusive breastfeeding or breast milk feeding via tube, spoon or paladai. Based on the available evidence of effectiveness, KMC will only be initiated in health facilities.[5]

### Comparison

The performance of the interventions will be assessed against a predefined success criterion of 80% or higher coverage of effective KMC at population level.

### Outcomes

The assumption that current coverage with KMC initiated at a health facility is low or negligible in the study population will be confirmed by a baseline assessment of facilities.

The primary outcome will be effective coverage of KMC, defined as at least 8 hours of skin-to-skin contact in combination with exclusive breastfeeding in the preceding 24 hours among infants with birth weight <2000 g. It will be measured:
► At discharge from the KMC facility (defined as one where staffing, infrastructure and mechanisms are present to initiate and maintain KMC).
► At 7 days after discharge, in the home.
  Secondary outcomes include:
► Population-level duration of KMC, that is, mean of how many days KMC was received by infants in the population with birth weight <2000 g during the neonatal period.
► Proportion of infants with birth weight <2000 g receiving any KMC (any duration in last 24 hours) at discharge, 7 days post-discharge and at 28 days of age.
► Proportion of infants with birth weight<2000 g exclusively breastfed (based on 24-hour recall) at discharge, 7 days post-discharge and at 28 days of age.
► Neonatal mortality rate (NMR) (including early NMR) in infants <2000 g.
► Cost of the KMC model (measured in some of the sites).

### Sample size

We assume that 3%–5% of neonates born in Ethiopia and India will be <2000 g and that it will not be possible to evaluate 20% of these newborns because of early death or loss to follow-up. A sample size of 310 newborns per site will allow us to estimate 80% coverage of KMC with site-specific absolute precision of±5%. All the study sites will have sufficient numbers of births of eligible infants to enrol at least 310 during the evaluation phase. We anticipate the follow-up of a total of at least 2170 newborns <2000 g during the evaluation period.

### Study sites

The study will be conducted in a total of seven sites in Ethiopia and in India. Ethiopia and India are among the countries in Africa and Asia with the highest proportion of LBW babies. Selection of sites within each country considered the quality of the proposals and the desire to reflect regional diversity.

### Ethiopia

In Ethiopia, newborn mortality remains high at 29 per 1000 live births.[9 10] The Ministry of Health has spearheaded a major campaign to increase facility births which appears effective.[11–15] Results from a recent survey (L10K) in 2015 conducted in 115 districts (woredas) in agrarian regions indicate facility births in the past 12 months to be around 53%.[11]

#### Amhara

The study will be conducted in two woredas of West Gojjam Zone, two woredas of South Gondar Zone, and Bahir Dar City Administration, Amhara Region. In West Gojjam Zone, the study includes one primary hospital. In

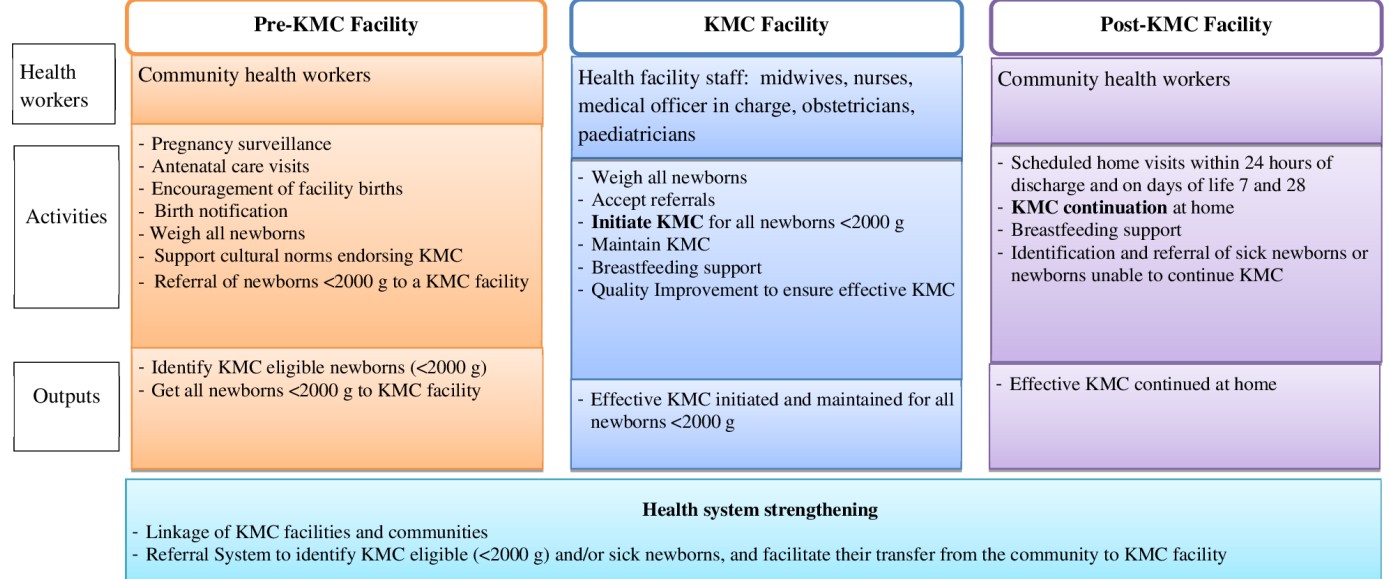

**Figure 1** Core framework for Kangaroo Mother Care implementation.

Bahir Dar City Administration, the study includes a large referral hospital, 1 primary hospital, 19 health centres, 42 health posts and their respective catchment populations. In South Gondar Zone, the study includes 1 general hospital and 1 primary hospital, 14 health centres and up to 59 health posts affiliated with these health centres, along with their respective catchment populations.

### Oromia—Addis Ababa
The Oromia—Addis Ababa site will implement the study in Akaki Kality sub-city in Addis Ababa and East Shoa Zone in Oromia region. Akaki Kality sub-city, one of the 10 sub-cities of Addis Ababa city administration shares border and culture with East Shoa Zone of Oromia region. The Akaki Kality sub-city has an estimated population of 180 000 living in 11 woredas in the sub-city.[14] Oromia Regional state has an estimated population of 27 million.[14] The study will be implemented in two woredas: Adami Tulu in the East Shewa Zone and Arsi Robe in the Arsi Zone.

### SNNPR
Southern Nations, Nationalities, and Peoples' Region (SNNPR) has a facility birth rate of 25.2%.[10] The sites proposed for study are Hawassa city administration, Shebedino woreda and Dale woreda, situated in Sidama Zone. Hawassa is the capital of SNNPR and its administration consists of eight sub-cities with urban and rural kebeles. The two public hospitals in Hawassa city—namely Hawassa University Comprehensive Specialised Hospital and Adare General Hospital—are included in the study. Leku is the capital of Shebedino woreda and consists of urban and rural kebeles. Leku Primary Hospital is the site for Shebedino woreda. Yirgalem is the capital of Dale woreda. Yirgalem General Hospital is the site for Dale woreda.

### Tigray
According to the Ethiopia Demographic and Health Survey (EDHS) 2016, in the Tigray region, 69% of the women who gave birth 2 years preceding the survey delivered in facilities. The study will be implemented in two zones, namely south-east zone and Mekelle special zone. In the south-east zone, two districts (Enderta and Degu-Tembien woredas) will be included in the study. In the selected districts and Mekelle Zone, there are 1 comprehensive referral hospital, 2 general hospitals, 1 primary hospital, 22 health centres and 5 private health facilities. All these health facilities will be included in the study.

### India
India accounts for one quarter of newborn deaths in the world. Neonatal deaths represent about 50% of under-five deaths and 70% of its infant deaths.[16] About 27% of all babies are LBW (<2500 g).[17–20]

### Haryana
The study will be implemented in one of Haryana's 22 districts (Sonepat). The LBW rate is 18% in Sonepat.[21 22] Around 5% of all infants weigh less than 2000 g at birth (*information from government records*). The institutional delivery rate is 80.5% and 90% of births are conducted by skilled birth attendants.[23] Infant mortality in Haryana is 36/1000, similar to the Indian average.[24] The NMR is 28/1000, a major proportion of infant mortality rate.[25]

### Karnataka
Infant mortality rate is 28/1000 and NMR is 22/1000 live births.[24 25] The research project will be conducted in the Koppal district, located 400 km away from Bangalore. This is an under-served, in the northern region of the state identified as a high priority district by the government.[26] The district has a LBW prevalence of about 25%

**Table 1** Description of study implementation teams

| | Research | | | Government implementation |
|---|---|---|---|---|
| | Programme learning team | Implementation support team | Outcome evaluation team | Health system team |
| Phase 1 | Conduct formative research to develop the initial KMC delivery model Conduct programme learning to understand how well the KMC delivery model is working and to redesign the model based on lessons learnt | Support government implementation of KMC | In the catchment area of the facilities involved in Phase 1, collect quantitative data on: ▶ Births in facilities and community ▶ Births <2000 g in facilities and community ▶ Number of newborns receiving KMC at time of discharge from health facility, 7 days post-discharge and at 28 days of age ▶ Duration of skin-to-skin care | Deliver and monitor the KMC intervention |
| Phase 2 | Monitor process indicators to learn how implementation of KMC delivery models happens in all study facilities and identify ways to refine the model based on context-specific findings | Assume an advisory role to support government implementation of KMC | In the entire study population, collect quantitative data as earlier | Deliver and monitor the KMC intervention |

KMC, Kangaroo Mother Care.

and a NMR of 42/1000. About 80% of births are institutional deliveries.

### Uttar Pradesh (UP)

UP is the most populous state in India and accounts for a quarter of India's burden of neonatal deaths.[25] UP's NMR of 37/1000 is among the highest in the country.[25] The prevalence of LBW in UP is 25%.[27] The study will be implemented in the Raebareli district with an institutional delivery rate of 93%. The study will cover the district hospital and ten community health centres, along with their catchment blocks.

### Ethical considerations

Since the intervention will be provided by the government as standard of care for LBW, there will be no request for consent from individuals to receive the intervention. Individual written informed consent will be requested from mothers, caregivers and health workers for the collection of information. For those unable to read, the informed consent form will be read out by the research team member before asking consent.

### Patient and public involvement

The question addressed by this research project was identified as a priority by the governments of Ethiopia and India. They participated in planning the study through the ministries of health. Patients and communities will play a major role in designing the models developed and tested, as described below under Phase 1—formative research. Results will be disseminated to the communities using channels available to the health system, as well as to the state and national health authorities, and through publication in peer-reviewed journals.

### Study approvals

Study approvals were obtained at the local, national and international levels.

### Study implementation strategy

#### Core framework for KMC implementation

KMC promotion will be implemented at three levels (figure 1):

'Pre-KMC facility'—the aim at this level is to maximise the proportion of newborns under 2000 g to be born in/brought to KMC implementing health facilities.

'KMC facility'—the aim at this level is early initiation of KMC and adherence to long duration (>8 hours) skin-to-skin care with exclusive breastfeeding within the facility.

'Post-KMC facility'—the aim at this level is continuation of effective KMC at home through the neonatal period or until the baby no longer accepts it.

#### Study teams

The KMC intervention will be delivered by health workers who are part of the health system. The research team's implementation role will be only supportive. This research team will be comprised of three small teams:

▶ The 'programme learning team', to conduct formative research and process monitoring, help formulate the intervention and process evaluation of implementation.

▶ The 'implementation support team' to guide and support the health system so that the environment supports KMC and health workers implementing the intervention.

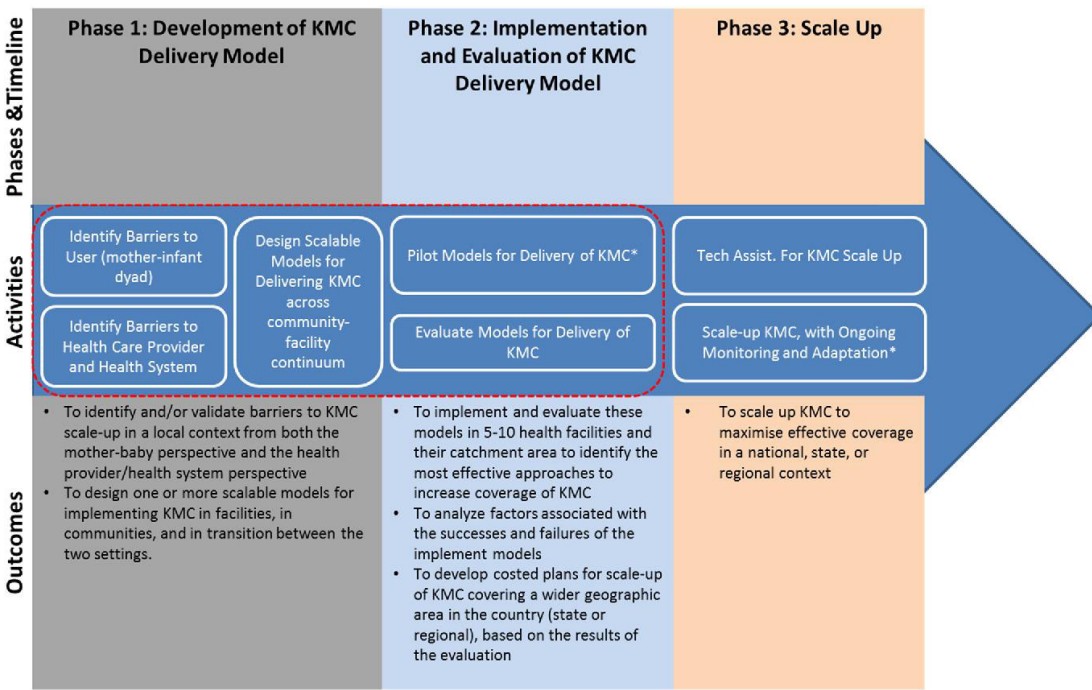

**Figure 2** Logic model for Kangaroo Mother Care scale-up.

► The 'outcome evaluation team' to measure the study outcomes in the population acting independently from the other two teams (table 1).

## Phases of implementation

The implementation will happen in two phases (figure 2).

### Phase 1: to develop a delivery model for KMC

#### KMC delivery model cycles of improvement

The initial delivery model will be developed based on the formative research and synthesis of literature. Programme learning from the delivery of this model and outcome measurements will immediately feed into a stage of model review and redesign/refinement, aimed at improving the coverage. We expect that at least three cycles of model development-implementation-evaluation-refinement will be needed until reaching a scalable high coverage

model. Each cycle may take 3–4 months. The first cycle may involve one facility but the second and third will be larger, including up to one-third of all study facilities.

At the end of each cycle, findings will be shared with an advisory group of stakeholders and their input will be sought to improve the model. New iterations of the model will be developed and further refined and reviewed with advisors. Figure 3 presents this iterative implementation-optimisation of the model.

### Formative research

The study's first step will be the programme learning research team undertaking formative research to identify and understand barriers and enablers to KMC in the specific context. We will follow the COM-B model to guide understanding of behaviour change in the implementation context, and the development of behavioural targets as a basis for designing interventions with users.[28] It proposes that people need capability (C: psychological or physical ability to enact a behaviour), opportunity (O: physical and social environment that enable a behaviour) and motivation (M: reflective and automatic mechanisms that activate or inhibit a behaviour) to perform a behaviour (B). Domains of the formative research are presented in table 2. Based on the knowledge gathered, we will develop an initial model of KMC implementation at the study site.

### Implementation of KMC test models

The initial model will be implemented in collaboration with facility and community workers with guidance from the implementation support team. Selection of facilities

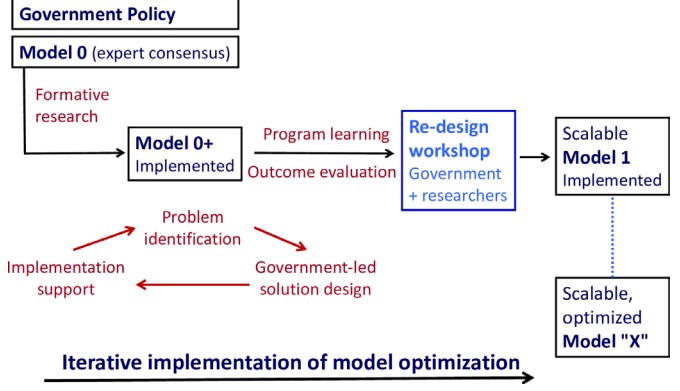

**Figure 3** Optimisation of implementation model.

**Table 2** Domains of enquiry for formative research

| Domains of enquiry | Activities |
| --- | --- |
| **KMC intervention**—to inform the design of the technical package of KMC | ► Review KMC scientific literature; existing government guidelines on KMC; experience on implementation of KMC<br>► Develop guidelines and protocols for health staff in facility and community healthcare workers<br>► Develop communication materials for communities, pregnant women and their families |
| **Users and implementers**—to understand the barriers and enablers to KMC adoption and adherence | **Mothers/families**<br>► Explore perceptions of mothers and family members around preterm babies, low birth weight, care-seeking, institutional deliveries, private facilities, perception of quality; understand and explore ideas to generate demand for KMC<br>**Health workers**<br>► Assess understanding and experience of KMC; understanding of birth weight, prematurity<br>► Assess knowledge, skill, time and motivation for newborn care |
| **Health systems**—to improve the delivery of KMC at the facility and community level | **Health facilities**<br>► Understand current KMC practices in health facilities; map behaviours that could impact adoption of KMC within these facilities<br>► Design a KMC facility readiness checklist based on local feasibility and provisions<br>► Assess physical infrastructure and resources available for KMC<br>**Link between community and facility**<br>► Assess current engagement between CHWs and health facilities and explore efficient process of engagement<br>► Assess referral system<br>**Assess data needs, measurement systems and analytics support and its compatibility and adaptability for KMC**<br>► Assess existing management information system; assess registries that exist at each level of the healthcare system<br>► Design and test electronic data management systems to guide and accelerate adoption and improvement of KMC<br>**Leadership and policy makers**<br>► Engage leadership and decision makers for their support of KMC<br>► Assess need of additional workers and current human resource capacity gap<br>**Other stakeholders**<br>► Map and engage all professional, organisational, health system, administrative and social networks that may impede or support KMC adoption and spread in the study area |
| **Context**—to develop adaptable delivery models | ► Understand perceptions of care of small babies and logistical challenges to care of high risk babies within the setting<br>► Explore motivations for facility versus home deliveries<br>► Map relationships between doctors and nurses, CHWs and mothers/families |

CHW, community health workers; KMC, Kangaroo Mother Care.

to implement KMC test models will be based on factors such as number of births, type of health workers, services delivered (eg, presence of newborn intensive care units or special care nurseries) so that they capture a major proportion of deliveries in the study district. Implementation will occur at three levels: pre-KMC facility, KMC facility and post-KMC facility (figure 1).

### Pre-KMC facility
To effectively implement KMC, eligible newborns need to be identified and cared for in facilities where KMC is initiated. All births should be identified, and babies weighed. Eligible babies, if born outside KMC facilities, will be referred to these facilities through strengthened government transport and linkage mechanisms. The pre-facility level includes efforts to: increase facility deliveries, expand early assessment of those delivered at home, so

that eligible newborns are referred to a KMC facility, and create a social norm around the practice of KMC.

### KMC facility
Facility level activities will aim to strengthen knowledge and skills of health providers and implement KMC protocols so that they provide high-quality care. All stable newborns would be placed in skin-to-skin contact and initiated breastfeeding within the first hour of life. In-born and referred newborns would be screened for KMC eligibility (<2000 g and stable). If meeting eligibility criteria, KMC would be initiated. If the newborn is sick, he/she will be stabilised first and then reassessed for KMC eligibility and initiation.

During the newborn's facility stay, nurses will do a KMC assessment following a checklist which examines skin-to-skin contact (number of hours) and breastfeeding,

 KMC Scale-Up Study Group. *BMJ Open* 2019;**9**:e025879. doi:10.1136/bmjopen-2018-025879

counselling on KMC, and problem-solving if barriers exist. Nurses will document the adoption of KMC. At time of discharge, mothers and family members will be counselled so that KMC is continued at home. Facility staff will notify community health workers (CHWs) about discharges and discuss how to support continued KMC at home.

### Post-KMC facility

After discharge, CHWs shall visit the home within 24 hours following the post-facility KMC protocol. CHWs will continue to visit the newborn following the standard national postnatal care visit schedule. At each visit, CHWs will perform a KMC assessment using the follow-up checklist including skin-to-skin duration and breastfeeding, counselling and problem solving. CHWs will identify and refer sick newborns to the facility and assist the referral.

### Health systems strengthening

Linkages between facility and community will be strengthened so that KMC is continued at home. This will be facilitated through establishment of communication systems between facilities and CHWs.

### Outcome measurement

The study outcome evaluation team will conduct an independent evaluation of the effectiveness of the model in achieving 80% or higher coverage with KMC among the target population. Data will be collected at health facilities and the community.

### Phase 2: implement and evaluate the chosen KMC delivery model

The most promising KMC delivery model will be chosen from Phase 1 based on programme learning findings and coverage. During Phase 2, the chosen model will be implemented in all selected facilities and data will be collected in the study area to measure the coverage with effective KMC.

### Implementation of the KMC delivery model

The model will include KMC guidelines, protocols, communication and training materials, and tools for successful implementation. The delivery model will be implemented by the health system in all study facilities and their catchment areas. The implementation support team will continue to be active but now playing only an advisory role to the government implementation teams. There will be ongoing mentorship and QI feedback cycles within each facility.

Implementation activities will include guideline dissemination, healthcare provider and CHW training, identification of LBWs, community awareness and demand generation, establishment of referral and supervision systems, monitoring, and continuous feedback loops. Key activities are further described below.

*Community awareness and demand generation:* Contacts between families and the health services will be utilised to promote awareness of KMC and of KMC units. Pregnant women and mothers of under-fives, for example, come in contact with health services during antenatal care, immunisation sessions and routine home visits by CHWs. Additional activities, such as village health promotion days, will be used. Messages developed will be communicated through posters, pamphlets, banners and wall paintings and disseminated at strategic locations in the community and at facilities with maternity wards, antenatal clinics, postnatal clinics, paediatrics wards in addition to the KMC wards. Audiovisual media such as local network for television channels or local radio may also be used.

Consistent messages from multiple channels, including interpersonal communication with health providers in public and private sectors and CHWs, will reinforce the information. Experience indicates that the endorsement from doctors seems critical to adoption of KMC. Orientation sessions will be conducted with all stakeholders and community representatives likely to influence community behaviour and norms.

*Identification of resources and staffing within the health system:* KMC is a new intervention for most health facilities in Ethiopia and India. Before initiating field work, discussions with national and local government officials will be held to promote buy-in. During these meetings, a landscaping overview of staffing, equipment and supplies at study health facilities will be completed. Together with government partners, research teams will meet with the leadership at study health facilities to introduce the KMC initiative, discuss their concerns and engage their involvement. We will jointly identify staff to participate in KMC implementation at each level of the health system.

*Development of intervention materials:* With Ministries of Health and country-specific teams, we will develop a standardised set of KMC protocols, guidelines, training materials, job tools and checklists to support implementation. Existing training manuals on newborn care in Ethiopia and India include KMC training. We will update them to include strategies to initiate and continue KMC at health facilities and plans for follow-up care at home. These study manuals will be used to train care providers in the selected study health facilities. To support health workers in implementing KMC, we will develop job-aids, checklists and data collection forms.

*Launch of the KMC implementation programme:* A 1 day workshop will be conducted involving officials and experts from study sites including regional and district health offices, health facilities and maternal, newborn and child health (MNCH) implementing partners. The workshop will orient stakeholders and strengthen local partnerships for successful implementation of project activities.

*Implementation team training:* We will provide an intensive training on KMC for healthcare workers from study facilities and CHWs from the community. Training will involve lectures and hands-on training, including technical updates, counselling skills, follow-up care at health facility and at home, and documentation of the KMC services. Trainers will be experienced local paediatricians and faculty from the site-specific institutions and the pool

**Table 3** Process indicators for KMC implementation

| Process | Indicator |
|---|---|
| Identification of pregnant women | ▶ Proportion of pregnant women identified by CHW in the antenatal period |
| Exposure to KMC priming interactions in the antenatal period | ▶ Proportion of identified pregnant women who received KMC information in the antenatal period from CHWs (either at home visits or during planned community interactions)<br>▶ Proportion of identified pregnant women who received KMC information during Antenatal Care (ANC) visits at health facilities |
| Delivery in KMC facility | ▶ Proportion of women who deliver in a KMC facility versus in another facility or at home |
| Weighed at birth | ▶ Proportion of live born babies who were weighed at birth, overall and by place of birth |
| Identification of newborns <2000 g | ▶ Proportion of babies identified as <2000 g, overall and by place of birth |
| Referral | ▶ Proportion of newborns <2000 g who were born at home or in non-KMC facility who were successfully referred to KMC facility for care |
| Initiation of KMC | ▶ Proportion of newborns <2000 g born in KMC facility who were initiated on KMC<br>▶ Proportion of newborns <2000 g referred to KMC facility who were initiated on KMC |
| KMC monitoring | ▶ Proportion of KMC initiated newborns monitored by facility staff according to protocol<br>▶ Proportion of facility KMC newborns discharged according to criteria; left against medical advice; referred out; died before discharge<br>▶ Duration of stay in facility |
| KMC follow-up | ▶ Proportion of newborns discharged from facility receiving KMC who received follow-up per protocol |

CHW, community health worker; KMC, Kangaroo Mother Care.

of trainers already available in the state. Training components will include advantages of KMC, appropriate KMC techniques, possible barriers and ways to address them, how to support mothers to initiate and sustain KMC and counselling mothers/family members at discharge. Practical training will be coordinated centrally, at a dedicated KMC unit, initially through simulators and subsequently with mothers and their babies. Refresher training sessions will be held. Follow-up with supportive supervision will be set up to ensure that the physicians and nurses translate their learning into implementation at their facilities. The supportive supervision will be initially ensured by the study team.

The CHWs and their supervisors will be trained on essential newborn care. Master trainers will be involved. The monthly meetings of the CHWs will provide opportunities for follow-up re-training and query-solving of problems encountered during their work in the community.

Immediately after training, objective structured clinical examinations will be conducted to test trainees' knowledge and skills. Mentors' training will be done at each level of the health system.

## Process monitoring

The programme learning team will transition to documenting implementation processes by measuring process indicators. Process indicators for KMC implementation were described in table 3.

## Outcome measurement

The outcome measurement team will evaluate the effectiveness of the model in achieving 80% or higher coverage with KMC among the entire study population as described earlier in the outcome measurement section.

## Evaluation

Effective KMC coverage is defined as the number of newborns receiving KMC divided by the total number of newborns eligible for KMC (<2000 g and stable) in the facilities and in the study population during the evaluation period.

To measure the denominator for effective KMC coverage it is preferred that all facility births be identified by the outcome measurement team during facility visits, review of birth registers, and/or calls to the facilities. In addition, the outcome measurement team will liaise with CHWs to obtain the number of home births and their birth weights.

To calculate the numerator for effective KMC coverage, the outcome measurement team will quantify the number of newborns under 2000 g who received KMC in the study population. All identified newborns under 2000 g will be followed by the outcome measurement team in the facility or at home to collect data at four points that is, day of initiation of KMC, day of discharge from the KMC facility, seven days post-discharge and at age 28 days. At each time point, the team will measure KMC based on records, and/or maternal report, if post-discharge.

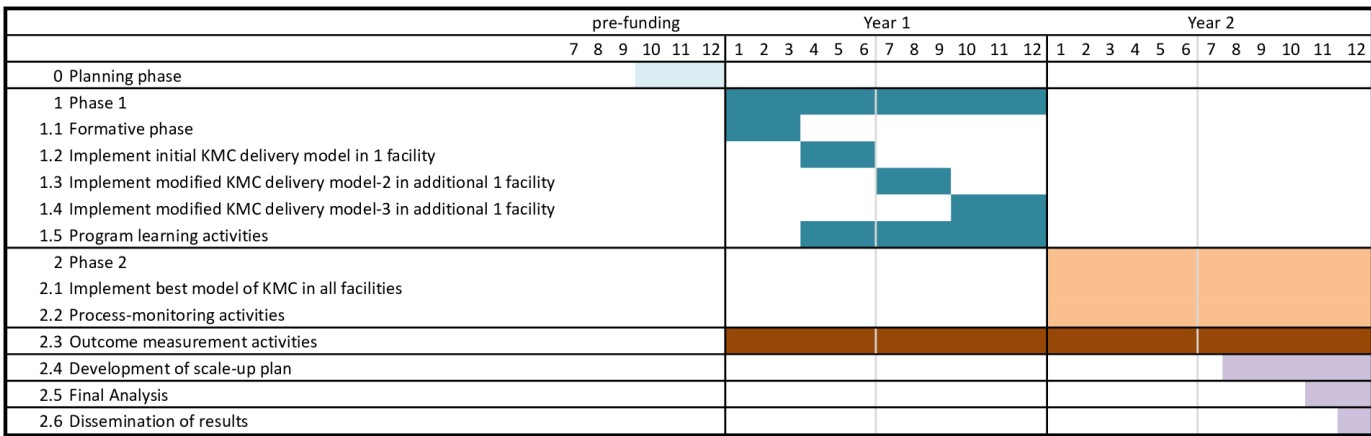

**Figure 4** Research timeline.

## Timeline

The study timeline is displayed in figure 4.

## Data management

Data collected will be linked to participant identification numbers. Each research team will be responsible for data management for its site. The cleaned data set will be sent to WHO at monthly intervals for review and feedback. Data quality assurance methods would include online real-time checks for missing values and inconsistency; random spot checks and data validation will be coupled with independent quarterly audits.

## Oversight

The Maternal, Newborn, Child and Adolescent Health Department of WHO, Geneva will create a Steering Committee and be responsible for organising its meetings. The Steering Committee will be the technical body responsible for the oversight.

## DISCUSSION

Despite its scientifically documented benefits, scaling up of KMC has proven an elusive goal for about 40 years. With increased awareness of the magnitude of newborn mortality and the clear interest of the governments of Ethiopia and India in making KMC available to all newborns of birth weight below 2000 g, it is timely to implement this project. It provides a great opportunity to weave KMC within the existing health system and, in collaboration with the government, to find effective models of implementation applicable at scale and sustainable. The findings from this implementation research project will provide inputs to policy makers to formulate KMC implementation guidance for state or nationwide scale-up, if one doesn't yet exist, or to improve existing ones. The findings will have implications for additional actions to strengthen provision of essential newborn care.

Literature on barriers and facilitators of KMC within health facilities is available.[29–31] This project aims to develop models and test them in cycles with an inbuilt iterative learning component and strong government-research partnerships that will ultimately lead to identification of effective models of scale-up in large countrywide settings. While we focus on KMC coverage in our outcomes, our aim is to develop a model that incorporates KMC as an integral component of newborn care so that the KMC scale-up is not projected as a vertical programme. The experiences from this research will not only inform us on the challenges to scale up KMC and ways to effectively address them but will have the potential to help the design and implementation strategy of future infant and child care programme in a way that their accelerated uptake by health system and the communities is strengthened. The findings of this implementation research will also be useful in other low/middle-income countries, besides India and Ethiopia.

**Acknowledgements**
We acknowledge the support of Dr. Suresh Kumar Dalpath, Deputy Director (Public Health Planning M&E SHSRC) Health Department, Haryana, India for his contributions to the development of the protocol, as well as the support of Anayda Portela, Daphne McRae and Bertrand Leduc in the preparation of the final manuscript.

**Collaborators** Araya Abrha Medhanyie,[1] Hibret Alemu,[2] Anteneh Asefa,[3] Selemawit Asfaw Beyene,[1] Fisseha Ashebir Gebregizabher,[4] Khalid Aziz,[5] Nita Bhandari,[6] Habtamu Beyene,[7] Thomas Brune,[8] Grace J Chan,[9,10] John N Cranmer,[11] Gary Lee Darmstadt,[12] Dereje Duguma,[13] Addisalem Fikre,[14] Bizuayehu Gashaw Andualem,[15] Abebe Gebremariam Gobezayehu,[16] Damen Haile Mariam,[14] Tedros Hailu Abay,[17] HL Mohan,[18] Arun Jadaun,[6] Krishnamurthy Jayanna,[20,21] F N U Kajal,[22] Arin Kar,[23] Raghav Krishna,[24] Aarti Kumar,[24] Vishwajeet Kumar,[24] Tarun Kumar Madhur,[6] Mulusew Lijalem Belew,[25] Rajini M,[26] Jose Carlos Martines,[27] Sarmila Mazumder,[6] Hajira Amin,[16] Prem K Mony,[28] Mekonnen Muleta,[29] Cynthia Pileggi-Castro,[30] Suman Rao,[31] Abiy Seifu Estifanos,[14] Lynn M Sibley,[32] Nalini Singhal,[33] Henok Tadele,[34] Abraham Tariku,[35] Ephrem Tekle Lemango,[36] Birkneh Tilahun Tadesse,[34] Ravi Prakash Upadhyay,[6] Bogale Worku,[14,37] Marta Yemane Hadush,[17] Rajiv Bahl.[30] [1]School of Public Health, Mekelle University College of Health Sciences, Mekelle, Ethiopia, [2]Urban Health, John Snow Inc, Addis Ababa, Ethiopia, [3]School of Public Health, Hawassa University College of Medicine and Health Sciences, Hawassa, Ethiopia, [4]Tigray Regional Health Bureau, Mekelle, Ethiopia, [5]Department of Neonatology, University of Alberta, Edmonton, Alberta, Canada, [6]Centre for Health Research and Development, Society for Applied Studies, New Delhi, India, [7]Southern Nations, Nationalities and Peoples' Regional Health Bureau, Hawassa, Ethiopia, [8]Department of Neonatology, Karloniska Institute, Calgary, Alberta, Canada, [9]Boston Children's Hospital, Department of Pediatrics, Harvard Medical School, Boston, Massachusetts, USA, [10]Department of Epidemiology, Harvard T.H. Chan School of Public Health, Boston, Massachusetts, USA, [11]Nell Hodgson Woodruff School of Nursing, Emory University, Atlanta, Georgia, USA, [12]Department of Pediatrics, Stanford University School of Medicine, Stanford, California, USA, [13]Oromia Regional Health Bureau, Addis Ababa, Ethiopia, [14]Addis Ababa University, School of Public Health, Addis Ababa, Ethiopia, [15]Amhara National Regional Health Bureau, Bahir

Dar, Ethiopia, [16]Emory Ethiopia, Addis Ababa, Ethiopia, [17]Department of Pediatrics and Child Health, Mekelle University College of Health Sciences, Mekelle, Ethiopia, [18]Community Mobilization, Karnataka Health Promotion Trust, Bangalore, India, [19]Centre for Health Research and Development, Society for Applied Studies, New Delhi, India, [20]Quality Improvement, Karnataka Health Promotion Trust, Bangalore, India, [21]Centre for Global Public Health, University of Manitoba, Winnipeg, Manitoba, Canada, [22]National Health Mission, Indian Administrative Service, Lucknow, India, [23]Karnataka Health Promotion Trust, Bangalore, India, [24]Global Health, Community Empowerment Lab, Lucknow, India, [25]Amhara Regional Office, Emory Ethiopia, Bahirdar, Ethiopia, [26]Department of Health and Family Welfare, Government of Karnataka, Bangalore, India, [27]Centre for Intervention Science in Maternal and Child Health, Universitetet i Bergen Senter for internasjonal helse, Bergen, Norway, [28]Division of Epidemiology and Population Health, St. John's Research Institute, St. John's National Academy of Health Sciences, Bangalore, India, [29]Private Consultant, Addis Ababa, Ethiopia, [30]Department of Maternal, Newborn, Child and Adolescent Health, World Health Organization, Geneva, Switzerland, [31]Department of Neonatology, St John's Medical College Hospital, Bangalore, India, [32]Global Health, Emory University School of Public Health, Atlanta, Georgia, USA, [33]Department of Neonatology, University of Calgary Cumming School of Medicine, Calgary, Alberta, Canada, [34]Department of Pediatrics and Child Health, College of Medicine and Health Sciences, Hawassa University, Hawassa, Ethiopia, [35]Federal Ministry of Health, Addis Ababa, Ethiopia, [36]Maternal and Child Health Directorate, Federal Ministry of Health, Addis Ababa, Ethiopia, [37]Pediatrics Society, Addis Ababa, Ethiopia.

**Contributors** Conceptualisation of the study and first drafting of the manuscript: AGG, MLB, LMS, GC, ASE, DHM, HT, BTT, KA, AAM, THA, SM, RU, NB, PKM, SPnR, KJ, ArK, AaK, GD, VK, RB, CP-C, JM. Subsequent revisions of manuscript drafts, completion of information on study settings and methods: JNC, AGG, MLB, BGA, HMA, LMS, GC, ASE, AF, HA, AT, DD, DHM, HT, BTT, AA, HB, BW, MM, TB, NS, KA, AAM, THA, SAB, FAG, MYH, ETL, SM, RU, TKM, AJ, NB, PKM, SPnR, KJ, ArK, MHL, RM, AaK, RK, FNUK, GD, VK, JM, RB. All authors read and approved the final manuscript.

**Funding** The World Health Organization, the sponsor, will fund the study with financial support from the Bill and Melinda Gates Foundation. WHO facilitated the preparation of the protocol, will monitor implementation and analysis of data. It will also facilitate the writing of the report and participate in the decision to submit the report for publication. The study teams will have ultimate authority over any of these activities.

**Disclaimer** The authors alone are responsible for the views expressed in this article and they do not necessarily represent the views, decisions or policies of the institutions with which they are affiliated.

**Competing interests** None declared.

**Patient consent for publication** Not required.

**Provenance and peer review** Not commissioned; externally peer reviewed.

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
