## [Reviewer comments · BMJ Open]

ARTICLE DETAILS

TITLE (PROVISIONAL)	Kangaroo Mother Care Implementation Research to Develop Models for Accelerating Scale-up in India and Ethiopia: Study Protocol for an Adequacy Evaluation
AUTHORS	Medhanyie, Araya; Alemu, Hilbret; Asefa, Anteneh; Beyene, Selemawit; Gebregizabher, Fisseha; Aziz, Khalid; Bhandari, Nita; Beyene, Habtamu; Brune, Thomas; Chan, Grace; Cranmer, John; Darmstadt, G; Duguma, Dereje; Fikre, Addisalem; Andualem, Bizuayehu; Gobezayehu, Abebe; Mariam, Damen; Abay, Tedros; Mohan, H L; Jadaun, Arun; Jayanna, K; Kajal, FNU; Kar, Arin; Krishna, Raghav; Kumar, Aarti; Kumar, Vishwajeet; Madhur, Tarun; Belew, Mulusew; M, Rajini; Martines, Jose; Mazumder, Sarmila; Amin, Hajira; Mony, Prem; Muleta, Mekonnen; Pileggi-Castro, Cynthia; Pn Rao, Suman; Estifanos, Abiy; Sibley, Lynn M.; Singhal, Nalini; Tadele, Henok; Tariku, Abraham; Lemango, Ephrem; Tadesse, Birkneh; Upadhyay, Ravi; Worku, Bogale; Hadush, Marta; Bahl, Rajiv

VERSION 1 – REVIEW

REVIEWER	Christina Tiyankhuleni Mathias University of KwaZulu-Natal, South Africa
REVIEW RETURNED	03-Dec-2018

GENERAL COMMENTS	Let a proof reader go through the manuscript and work on word spacing, otherwise the protocol is well structured and it is rich with current literature on low birth weight infants and Kangaroo Mother Care.
---

REVIEWER	David Ellard Warwick Clinical Trials Unit The University of Warwick United Kingdom
REVIEW RETURNED	04-Feb-2019

GENERAL COMMENTS	KANGAROO MOTHER CARE IMPLEMENTATION RESEARCH TO DEVELOP MODELS FOR ACCELERATING SCALE-UP IN INDIA AND ETHIOPIA: STUDY PROTOCOL FOR AN ADEQUACY EVALUATION Thank you for inviting me to review this interesting protocol. It is interesting to be reviewing a piece of implementation research. This is a large complex study involving a lot of people. Background information is good. I would have preferred to see the primary objective be called the aim or the research question with the objectives being how this will be addressed. I think the sub-
--

	objectives A & B are additional aims. This I think confuses the design a little. There is a good description of the research sites but no indication why or how they were chosen. The study teams section notes that it is NOT the researchers who are delivering the intervention but it is also clear the intervention is not yet developed. This brings me to my main worry at present I do not feel there is enough information on what is being done when and by whom. Timelines would be helpful when is all this happening? How long is phase one? How much time is allocated to develop the intervention once the formative work is done. Then one assumes that training has to be developed to 'train' the teams who are implementing this – what are the timeframes for this? The study does not seem to be based on any theoretical model or underpinning. QI is mentioned in the objectives but I see nothing that tells me how this will be done? I feel that for such a complex study I am not being given all of the methods. I again thank you for inviting me to review this article I wish you well with this interesting and valuable project and your future work.
--	---

VERSION 1 – AUTHOR RESPONSE

REVIEWER 1

Let a proof reader go through the manuscript and work on word spacing, otherwise the protocol is well structured and it is rich with current literature on low birth weight infants and Kangaroo Mother Care.

Response: We thank the reviewer for the very positive comments on the structure of the protocol and the review of the literature. We have removed the right-margin justification in the formatting and this has substantially reduce the problem with word-spacing. In addition, we have proof-read and eliminated further spacing problems.

REVIEWER 2

1. Thank you for inviting me to review this interesting protocol. It is interesting to be reviewing a piece of implementation research.

This is a large complex study involving a lot of people.
Background information is good.

Response: We thank the reviewer for the positive comments on the interest of this implementation research protocol and the background information that it presents.

2. I would have preferred to see the primary objective be called the aim or the research question with the objectives being how this will be addressed. I think the sub-objectives A & B are additional aims. This I think confuses the design a little.

Response: We implemented the reviewer's suggestion and, in doing so, also removed the presentation of the secondary objectives as they are later repeated in the section on secondary outcomes.

3. There is a good description of the research sites but no indication why or how they were chosen.

Response: We agreed that this information would be useful and added it to the first paragraph of the section on study sites.

4. The study teams section notes that it is NOT the researchers who are delivering the intervention but it is also clear the intervention is not yet developed. This brings me to my main worry at present I do not feel there is enough information on what is being done when and by whom.

Response: The reviewer is correct in observing that the investigators are not in charge of the delivery of the intervention. From its conceptualization, the study aimed to work with the government in each of the countries to develop a model that would allow government services to reach a high population coverage with the provision of KMC. The reviewer's comment makes us realize that in our attempt to be succinct in the presentation of the study we left out important details about the intervention. We have expanded the information provided in the section describing phase 2 (under Implementation of the KMC Delivery Model). We have deleted redundant information provided under a section titled "field logistics".

We hope that this will be found to be satisfactory.

5. Timelines would be helpful when is all this happening?
How long is phase one? How much time is allocated to develop the intervention once the formative work is done. Then one assumes that training has to be developed to 'train' the teams who are implementing this – what are the timeframes for this?

Response: Thank you for the important comment. We have now added a section with a figure (Figure 4) presenting the requested information on timeline.

6. The study does not seem to be based on any theoretical model or underpinning. QI is mentioned in the objectives but I see nothing that tells me how this will be done? I feel that for such a complex study I am not being given all of the methods.

Response: We followed the COM-B model. It guides both the understanding of behaviour change in the implementation context, and the development of behavioural targets as a basis for designing interventions with users [Michie, S., M.M. van Stralen, and R. West. The behaviour change wheel: a new method for characterising and designing behaviour change interventions. *Implement Sci*, 2011. 6: p. 42]. It proposes that people need capability (C: psychological or physical ability to enact a behaviour), opportunity (O: physical and social environment that enable a behaviour) and motivation (M: reflective and automatic mechanisms that activate or inhibit a behaviour) to perform a behaviour (B). We have also included a new figure (Figure 3) to present the process of iterative implementation of model optimization. We apologize for the limited information presented on how QI would be done. We would be happy to include details on it and further information on the theoretical model underpinning the study intervention but fear exceeding the length authorized. We would appreciate guidance from the reviewer and the editor.

7. I again thank you for inviting me to review this article I wish you well with this interesting and valuable project and your future work

Response: We thank the reviewer for the positive and concrete comments and for the good wishes